# Amine Plasma-Polymerization of 3D Polycaprolactone/β-Tricalcium Phosphate Scaffold to Improving Osteogenic Differentiation In Vitro

**DOI:** 10.3390/ma15010366

**Published:** 2022-01-04

**Authors:** Hee-Yeon Kim, Byung-Hoon Kim, Myung-Sun Kim

**Affiliations:** 1BioMedical Sciences Graduate Program (BMSGP), Chonnam National University, Hwasun 58128, Korea; khy9787@hanmail.net; 2Department of Dental Materials, College of Dentistry, Chosun University, Gwangju 61452, Korea; 3Department of Orthopaedic Surgery, College of Medicine, Chonnam National University, Gwangju 61469, Korea

**Keywords:** bone tissue engineering, 3D printing, bone regeneration, amine plasma-polymerization, plasma surface modification

## Abstract

This study aims to investigate the surface characterization and pre-osteoblast biological behaviors on the three-dimensional (3D) poly(ε-caprolactone)/β-tricalcium phosphate (β-TCP) scaffold modified by amine plasma-polymerization. The 3D PCL scaffolds were fabricated using fused deposition modeling (FDM) 3D printing. To improve the pre-osteoblast bioactivity, the 3D PCL scaffold was modified by adding β-TCP nanoparticles, and then scaffold surfaces were modified by amine plasma-polymerization using monomer allylamine (AA) and 1,2-diaminocyclohexane (DACH). After the plasma-polymerization of PCL/β-TCP, surface characterizations such as contact angle, AFM, XRD, and FTIR were evaluated. In addition, mechanical strength was measured by UTM. The pre-osteoblast bioactivities were evaluated by focal adhesion and cell proliferation. Osteogenic differentiation was investigated by ALP activity, Alizarin red staining, and Western blot. Plasma-polymerization induced the increase in hydrophilicity of the surface of the 3D PCL/β-TCP scaffold due to the deposition of amine polymeric thin film on the scaffold surface. Focal adhesion and proliferation of pre-osteoblast improved, and osteogenic differentiation was increased. These results indicated that 3D PCL/β-TCP scaffolds treated with DACH plasma-polymerization showed the highest bioactivity compared to the other samples. We suggest that 3D PCL/β-TCP scaffolds treated with DACH and AA plasma-polymerization can be used as a promising candidate for osteoblast differentiation of pre-osteoblast.

## 1. Introduction

Large bone defects are on the increase due to tumor, trauma, or infections [1]. The treatment options for the restoration of large bone defects are limited and remain a significant challenge in clinical trials because they have no ability to regenerate automatically [2].

Therefore, surgical operation is required for bone reconstruction. Bone autografts and allografts are considered ideal options for bone reconstruction, but many potential drawbacks of these treatment strategies, such as morbidity of donor site, limited ability for bone repair, and risk of infection, often limit clinical application [3,4,5].

From these points of view, bone scaffold was required for proper bone substitute of large bone defect repair [6]. Especially for large bone defects, bone scaffolds need to be rigid enough to support mechanical strength and pressure [7].

Recently, there is increasing interest in developing processes for the production of three-dimensional (3D) scaffolds optimized for bone tissue engineering applications. Three-dimensional printing technology has attracted attention resulting in advantages such as design and fabrication of the scaffold architecture’s internal structure, shape, porosity, pore size, and pore interconnectivity and external shapes [8,9,10]. 

Various biomaterials have been investigated as scaffold materials for repair of damaged bone tissue, including metals, ceramics, polymers (natural and synthetic), or their combinations [11]. Since bioceramics have similar chemical and structural properties compared to the mineral phase of human bones, they have been extensively studied as a biocompatible and osteoconductive material for bone regeneration [12,13]. However, their physicochemical properties, such as slow degradation rate in the human body and pure processability, limit application in bone regeneration [13]. 

Since biodegradable PCL polymers and their copolymers have excellent physicochemical properties such as biodegradability, biocompatibility, mechanical strength, and ease of fabrication, they have been used extensively as a scaffold material for bone tissue engineering [14]. 

To achieve success in bone tissue engineering, scaffold should be suitable for interactions with cells, such as with cell adhesion, migration, proliferation, and differentiation [15,16]. Even though biopolymers in bulk state have excellent physicochemical properties, their surface must be met with requirements for cell interactions. For example, most biopolymers have hydrophobicity due to a hydrocarbon chain backbone, and they have a limited biological cell response. From these points of view, surface modification techniques of biopolymer are required to enhance the cell–material interactions. 

Various techniques for surface modifications of the scaffolds have been investigated [17,18,19]. For example, chemical modification of scaffold surfaces has been tried by introducing functional groups or covalent coupling of bioactive molecules onto the surface, to endow bioactivity or hydrophilicity [17]. Surface modification of scaffolds using plasma irradiation have been also investigated to modify surfaces chemical properties and morphologies as it is regarded as an ideal tool for modification of complex-shaped surfaces to homogeneous surfaces, and furthermore it is a solvent-free technology [20]. Notably, plasma techniques are used to modify surface characteristics for better biocompatibility without changes in the bulk properties of scaffolds. 

Plasma-polymerization is a deposition technique where various organic monomers are introduced into a plasma discharge zone and converted into reactive fragments and polymer thin films (100Å–1 μm) are deposited [20,21,22]. These polymeric thin films are highly cross-linked and pinhole-free, resulting in their being insoluble, thermally stable, and chemically inert. Among various precursors, amine plasma polymers are considered as a useful reactive platform for immobilization of biologically active molecules such as proteins, collagen, DNA, and peptides via covalent coupling [21]. To create amine-group-decorated surfaces, plasma-polymerization using amine-based monomers is thought to be an efficient way.

For example, Barry et al. reported that allylamine plasma-polymerization induces higher N-concentrations on the outer and inner surfaces of PLA scaffold compared to the plasma grafting process. They also demonstrated that plasma deposition-treated scaffolds resulted in higher metabolic activity than that of grafted one [23]. With consideration of their results, plasma-polymerization is considered as a suitable technique for the introduction of an N element on bone scaffold surface. 

Liu et al. demonstrated that allylamine among various monomers is an excellent candidate to promote attachment, spreading, and proliferation of hASCs, as well as promotion of the osteogenic differentiation of them [24]. 

Song et al. reported the surface modification of coronary artery stents by plasma-polymerization using 1, 2-diaminocyclohexane (DACH) followed by chemical grafting of α-lipoic acid (ALA). The ALA-grafted polymeric film demonstrated good mechanical stability and blood compatibility [25].

Shin et al. performed the DACH plasma-polymerization to immobilize the bone morphogenic protein-2 (BMP-2) on a titanium (Ti) surface. The BMP-2 immobilized Ti surface demonstrated the higher osteoblast differentiation compared to a pristine Ti surface [26]. 

Many works have been studied on the amino-functionalized surface by amine-based plasma-polymerization for biomedical applications. However, there are few studies on the bioactivity of an amino-functionalized surface modified by DACH plasma-polymerization.

Therefore, we fabricated the 3D PCL scaffolds using an FDM 3D printing technique. In addition, 3D PCL scaffold was incorporated with β-TCP powder to improve the osteoconductivity, and carried out amine plasma-polymerization using AA and DACH monomer on 3D PCL/β-TCP scaffold in order to improve the pre-osteoblast adhesion, proliferation, and osteogenic differentiation.

## 2. Materials and Methods

### 2.1. Materials

PCL (average molecular weight = 45,000 g/mol), β-TCP (nano powder), and 3-(4,5-Dimethylthiazol-2-yl)-2,5-diphenyltetrazolium bromide (MTT) were purchased from Sigma-Aldrich (St. Louis, MO, USA). Allylamine (AA) and 1,2-diaminocyclohexane (DACH) used as monomer were purchased from Sigma-Aldrich (St. Louis, MO, USA). 

Minimum essential medium alpha (α-MEM) and trypsin-ethylenediaminetetraacetic acid (EDTA) were purchased from Gibco (Grand Island, NY, USA). Dulbecco’s phosphate buffered saline (DPBS) was purchased from iNtRON Bio (Sungnam, Korea). The osteogenic medium used in this study comprised of α-MEM containing 10% fetal bovine serum (FBS, Gibco, Grand Island, NY, USA), 1% penicillin-streptomycin (Gibco, Grand Island, NY, USA), 10 mM β-glycerol phosphate disodium salt hydrate (Sigma-Aldrich, St. Louis, MO, USA), and 50μg/mL ascorbic acid (Sigma-Aldrich, St. Louis, MO, USA). Deionized water was produced by an ultrapure water system (Arium^®^ro; Sartorius, Göttingen, Germany). The fluorescence staining reagents used in this study are as follows; 4% paraformaldehyde (T&I, Gwangju, Korea), triton X-100(Sigma-Aldrich, St. Louis, MO, USA), bovine serum albumin (Sigma-Aldrich, St. Louis, MO, USA). Anti-vinculin antibody and anti-paxillin antibody were purchased from Sigma-Aldrich (St. Louis, MO, USA). Antifade mounting medium used in this study comprised of for fluorescence were VECTASHIELD HardSet Antifade Mounting Medium with DAPI and VECTASHIELD HardSet Antifade Mounting Medium with phalloidin (Vector Laboratories, Burlingame, CA, USA). All reagents and chemicals in this study were used without any further purification.

### 2.2. Fabrication of 3D PCL and PCL/β-TCP Scaffolds and PCL Films

Three-dimensional PCL and PCL/β-TCP scaffolds were fabricated using Bio-Extruder equipment (3D Bio Printer, M4T-100, M4T, Daegu, Korea). Briefly, PCL pellets melted at 70 °C in a heating cylinder was mixed with 10 wt% β-TCP. PCL and PCL/β-TCP mixtures were ejected through heated nozzle of compressed dry air at a pressure of 640 kPa, and the feed rate was set to 240 mm/min. These scaffold struts could be plotted as layer-by-layer deposition on a stage. For in vitro study, scaffolds (strut size = 300 μm and pore size = 300 μm) were fabricated in to disk shape with a diameter of 20 mm and height of 1.2 mm (Figure 1(A-1,A-2)). In addition, scaffold sample was fabricated in to cylinder shape with diameter of 10 mm and height of 10 mm to measuring the mechanical strength test (Figure 1(B-1,B-2)).

To use in the surface analysis, PCL and PCL/β-TCP films were prepared by mixing PCL pellets with 10 wt% β-TCP powder in glass container at 70 °C. The polymer mixture solution was cast onto a glass substrate and then maintained until room temperature. The films with a thickness of 1 mm were cut into specimen size of 10 mm × 10 mm for the surface characterization.

### 2.3. Amine Plasma-Polymerization on the 3D Scaffolds

The amine plasma-polymerization of PCL and PCL/β-TCP 3D scaffold was carried out by radio frequency (RF = 13.56 MHz) capacitively-coupled plasma (Miniplasma Station, Daejeon, Korea) using monomer such as AA and DACH. 

The amine plasma-polymerization process was performed under the following conditions (Table 1):

Pre-treatment was performed to activate the surface of PCL/β-TCP 3D scaffold using argon plasma treatment. Moreover, post-treatment was performed to increase the degree of polymerization for amine polymeric thin film.

In order to perform the homogeneous amine plasma-polymerization to the interior structure of 3D PCL/β-TCP scaffold, we carried out amine plasma-polymerization the upper surface of the 3D PCL/β-TCP scaffold, and then turned it over again to perform amine plasma-polymerization.

### 2.4. Surface Characterization of 3D Scaffolds

The morphology of 3D scaffolds was observed using field emission scanning electron microscopy (FE-SEM, JSM-7500F+EDS, Oxford) with an energy dispersive spectrometry (EDS, Oxford Instruments, UK) detector. All samples were coated with gold sputtering for FE-SEM. The observation was carried out at an accelerating voltage of 15 kV at magnifications 50×. The beam intensity was 2.5 mA and the point EDS analysis were performed with a 60 s counting time at each point. 

The structural analysis of pure PCL, PCL/β-TCP, and β-TCP particles was carried out with X-ray diffractometer (XRD, SmartLab, Rigaku, Japan) using Cu-Kα radiation at 45 kV and 200 mA at room temperature. The XRD data were collected over 2θ range 10–50° at a step size of 0.02°/s.

Atomic force microscopy (AFM, XE-100, Park systems, Suwon, Korea) was used to analyze the surface topologies and roughness of 3D scaffolds before and after amine plasma-polymerization under non-contact mode with a scan rate of 0.1 Hz. AFM images were analyzed with XEI software. The Scan areas of 5 μm × 5 μm and 10 μm × 10 μm were randomly selected from the scaffold surface. An arithmetic mean of the root mean square (RMS) and roughness (Rq) was calculated directly from the AFM images.

Fourier transform infrared (FT-IR, UART Two, PerkinElmer, Akron, OH, USA) spectroscopy was used to characterize the chemical group of scaffolds after plasma-polymerization. The FTIR spectra were measured with a resolution of 4 cm^−1^ and with an accumulation of 4 scans in the range of 4000–400 cm^−1^. 

To evaluate the surface hydrophilicity, the degree of water spreading on the PCL and PCL/β-TCP film surface before and after amine plasma-polymerization was measured to compare water contact angle. The hydrophilicity of samples was examined by the sessile drop method using a goniometer (GS, Surface Tech, Gwangju, Korea). A water droplet (5 μL) was dropped by a syringe mounted vertically against the PCL and PCL/β-TCP films surface. After the water was applied on the surface for 5 s, the droplet arc and the angle of contact (θ) at the interface were measured and recorded by a charge-coupled device (CCD) camera. 

### 2.5. Mechanical Strength Evaluation of 3D Scaffolds

For the evaluation of mechanical strength, the compressive strength and the compressive modulus were measured using a 10.0 kN load cell universal testing machine (UTM, AG-X, Shimadzu, Kyoto, Japan). The cross-head speed of compression test was fixed at 0.5 mm/min. All experiments were performed in triplicate (*n* = 3).

### 2.6. In Vitro Pre-Osteoblast Evaluation

#### 2.6.1. Cell Culture 

The MC3T3-E1 (newborn-mouse-derived calvaria, ATCC CRL-2593) cells were cultured in α-MEM (α-MEM, Gibco, Grand Island, NY, USA) supplemented with 10% FBS, 1% penicillin incubated at 37 °C in a saturated humid atmosphere containing 5% CO_2_ and 95% air. The medium was replaced at every 3 days until the cells reached 90% confluence.

#### 2.6.2. Cell Seeding into 3D Scaffolds

Before cell seeding, 3D scaffolds were treated with 70% ethanol for 5 min in order to remove potential residues from the sample preparation. After, these were dried overnight at ambient temperature and were sterilized by ultra violet C light for 1 h. The scaffolds were washed three times with phosphate-buffered saline (PBS, 0.01 M, pH 7.4). Finally, MC3T3-E1 cells were seeded onto these scaffolds.

Finally, the sterilized 3D scaffolds were placed in 12-well culture plates and MC3T3-E1 cells were seeded onto these scaffolds, avoiding contact with the sides of the wells to improve seeding efficiency. The cells were incubated at 37 °C for 3 h and each of the wells was filled with 1 mL of fresh medium. For osteogenic differentiation studies, the osteogenic induction medium containing 0.1 μM dexamethasone, 10 mM sodium β-glycerophosphate, and 0.05 mM ascorbic acid-2-phosphate.

#### 2.6.3. Cell Proliferation

The proliferation of the MC3T3-E1 cells on scaffolds was determined the 3-(4,5-dimethylthiazol-2-yl)-2,5-diphenyltetrazolium bromide (MTT, Sigma-Aldrich) assay. On the scaffolds, 1 × 10^5^ cells/well were seeded and incubated for 1, 3, and 5 days.

At each of the sampling days, MTT solution was added to the well and incubated at 37 °C for 4 h for MTT formazan crystal formation. After developing the purple formazan colors by the reaction between metabolically active cells and tetrazolium salt, the medium and MTT solution were replaced with dimethyl sulfoxide (DMSO) to dissolve purple formazan. The absorbance in each well was then measured using an EPOCH, absorbance microplate reader (BioTek Instruments, Winooski, VT, USA) at 540 nm.

#### 2.6.4. Cell Viability

The cell viability was evaluated by staining cells using a live/dead cell staining kit (Biovision, Milpitas, CA, USA). The MC3T3-E1 cells were seeded on the scaffolds at a density of 5 × 10^5^ cells/well on scaffolds in 12-well plates and were cultured on scaffolds for two days in CO_2_ incubator. 

After two days, the culture medium was removed from the scaffolds and the scaffolds were washed twice with PBS. Then, 1 mL of the staining solution (1 mM cell-permeable green fluorescent dye and 2.5 mg/mL of propidium iodide) per well and the culture plates were returned to the incubator for 20 min. Live (green) and dead cells (red) were imaged with a fluorescence microscope (NI-SS, Nikon, Tokyo, Japan).

#### 2.6.5. Cell Focal Adhesion

To investigate cell behavior, we observed the focal adhesion of MC3T3-E1 cells cultured in α-MEM medium for 5 h on the scaffolds.

Staining actin filaments with phalloidin can detect the actin cytoskeleton of MC3T3-E1 cells on the 3D scaffolds. For actin cytoskeleton and focal adhesion identification, 1 × 10^5^ cells/well were seeded on the scaffolds and incubated for 5 h to adhere to each surface of scaffold. The MC3T3-E1 cells on the scaffolds were washed in PBS and fixed in 4% PFA for 15 min. The fixed cells were permeabilized with 0.1% Triton X-100, blocked with 1% BSA for 30 min and incubated overnight at 4 °C with anti-vinculin antibody and anti-paxillin antibody diluted 1:50. After overnight, the cells on the scaffolds were washed thrice with PBS and incubated with secondary antibody conjugated to Alexa Fluor 488 (1:1000) for 2 h. Then, the scaffolds were washed in PBS and mounted in mounting medium with DAPI and phalloidin mixed at 1:1. Fluorescence of vinculin and paxillin (green), actin fibers (red), and nuclei (blue) were imaged with a fluorescence microscope (NI-SS, Nikon, Tokyo, Japan).

#### 2.6.6. Cell Differentiation

By measuring the ALP (Alkaline phosphatase) activity and ALP staining, a biochemical marker of osteoblasts, osteogenic differentiation capacity was evaluated.

All the PCL and PCL/β-TCP were placed in a 12-well plate and seeded with a density of 1 × 10^5^ cells/well. After culturing the MC3T3-E1 cells in osteogenic differentiation media for 7 and 14 days, ALP was determined by quantifying the release of p-nitrophenol (p-NP) from p-nitrophenyl phosphate (p-NPP). On the 7th and 14th day of differentiation, the scaffolds cultured with MC3T3-E1 cells were gently rinsed twice with PBS. The cells were lysed in RIPA buffer containing Xpert Protease Inhibitor Cocktail (100×) and Xpert Phosphatase Inhibitor Cocktail (100×) (Gen-Depot, Barker, TX, USA) for 15 min on ice. The lysate was centrifuged at 2500× *g* for 10 min at 4 °C and the clear supernatant was incubated with p-nitrophenyl phosphate (p-NPP) solution for 30 min at 37 °C. The reaction was stopped by adding 600 μL of 1.0 M NaOH. The ALP activity was determined by measuring the absorbance at 405 nm using an EPOCH, absorbance microplate reader (BioTek Instruments, Winooski, VT, USA) and normalized to the protein concentration. The protein concentration was determined by BCA protein assay (Pierce, Rockford, IL, USA). The data are expressed as μmole p-NP/min/μg protein.

Alkaline phosphatase (ALP) staining was carried out using an Alkaline phosphatase Detection Kit (SCR004, Millipore, Burlington, MA, USA) as per manufacturer’s instructions. On the 7th and 14th day of differentiation, the scaffolds cultured with MC3T3-E1 cells were washed with PBS twice and then fixed with 4.0% formaldehyde for 2 min. The cells were rinsed with TBST (20 mM Tris-HCl, pH 7.4, 0.15M NaCl, 0.05% Tween-20). The cells were stained with stain solution for 15 min at room temperature and rinsed with TBST twice. The images of the stained cells were captured using a digital camera.

#### 2.6.7. Bone Mineralization (Alizarin Red Staining) 

Bone mineralization of the cells cultured in osteogenic differentiation media on 3D scaffolds was evaluated by alizarin red staining. On the 7th and 14th day of differentiation, the cells were washed with PBS twice and then fixed with 4% formaldehyde for 15 min. The cells were stained with alizarin red staining solution (pH = 4.2) for 20 min at room temperature and rinsed with PBS twice. The images of the stained cells were captured using a digital camera. 

To analyze the calcium content of the calcified nodules quantitatively, scaffolds stained with alizarin red were bleached with diluted 10% (*w*/*v*) cetylpyridinium chloride in 10 mM sodium phosphate (pH = 7.0) at 37 °C for 15 min. The supernatant was spectrophotometrically analyzed at 560 nm using a NanoDrop spectrophotometer (BioTek, Winooski, VT, USA). These experiments were performed in triplicate.

#### 2.6.8. Western Blotting

To investigate osteogenic protein expressions, Western blotting of MC3T3-E1 cells was carried out as follows: Cells were cultured on 3D PCL scaffolds and 3D PCL/β-TCP scaffolds before and after amine plasma-polymerization for 7 and 14 days. After that, cells on the scaffolds were washed with PBS twice. The cells were lysed in RIPA buffer containing Xpert Protease Inhibitor Cocktail (100×) and Xpert Phosphatase Inhibitor Cocktail (100×) (Gen-Depot, Barker, TX, USA). This solution was centrifuged for 10 min at 4 °C (13,000 rpm). The cell lysates (supernatant) were then used to measure protein concentration using the BCA Protein Assay kit (Pierce, Rockford, IL, USA). Proteins (30 μg) were loaded into 7.5–15% sodium dodecyl sulfate polyacrylamide gel electrophoresis (SDS-PAGE), transferred to a polyvinyl difluoride (PVDF) membrane. Following transfer, membranes were blocked with 5% skim milk in TBS-T for 1 h, probed with an appropriate primary antibody against ALP (1:500), RUNX-2 (1:500), OPN (1:5000), and β-actin (1:10,000), respectively. Then, the membranes were incubated with anti-mouse and anti-rabbit secondary antibody (1:5000) for 2 h at room temperature. The immunoblot bands were detected by chemiluminescence (ECL, Pierce, Rockford, IL, USA) and then quantified with digital analyses using the ImageJ software program.

### 2.7. Statistical Analysis

The results of the in vitro evaluations were analyzed statistically using the one-way analysis of variance (ANOVA) combined with Tukey’s multiple comparison test. The statistical significance was evaluated at ** *p* < 0.01 and * *p* < 0.05. All data represent the mean ± standard error of the mean. All statistical analyses were performed using IBMSPSS Statistics for Windows version 23.0 (IBM Corp., Armonk, NY, USA).

## 3. Results

### 3.1. Surface Characterization of Scaffold

Figure 2(A-1,B-1) shows the FE-SEM images of 3D PCL and PCL/β-TCP scaffolds with 0°/45° strut layout pattern and 300 μm pore size, respectively. From this scaffold images, we could identify well deposited by 3D printing using a PCL and PCL/β-TCP materials. 

In Figure 2(B-2), it was observed that β-TCP nanoparticles of the PCL/β-TCP scaffolds were randomly distributed on the scaffold surface. The EDS analysis of the PCL/β-TCP scaffold demonstrated the presence of Ca, P, and O, which is consistent with β-TCP, as shown in Figure 2(B-3). These results indicate the β-TCP nanoparticles well incorporate to PCL scaffold.

To identify the crystalline phases of PCL, β-TCP, and PCL/β-TCP, XRD analysis was performed. In order to identify the incorporated β-TCP in PCL scaffold, β-TCP powder was used in XRD analysis. The XRD patterns of PCL, PCL/β-TCP, and β-TCP are presented in Figure 3. The PCL showed two strong peaks located at 2θ = 21.3° and 23.8°, which were respectively associated with the (110) and (200) reflections of a polyethylene-like crystal structure [27]. The β-TCP show diffraction patterns that are consistent with the β-TCP reference (JCPDS file no. 09-169, International Center for Diffraction data). The PCL/β-TCP scaffold showed PCL peaks and the peaks of the β-TCP particle. From this result, we proved that the β-TCP particles were well- incorporated in PCL scaffolds.

Figure 4 shows the AFM images of the four different samples and their Rq from two different scan sizes, respectively. The pristine PCL scaffolds showed relatively smooth surfaces, as shown in Figure 4(Aa) 2.645nm and Figure 4(Ae) 3.580nm. On the other hands, the Rq of PCL/β-TCP was significantly increased and observed the rough surface compared to PCL surface, because the incorporation of β-TCP particle into PCL polymer (Figure 4(Ab,Af)). The PCL/β-TCP scaffolds after 1 min of AA and DACH plasma-polymerization showed similar surface roughness compared to before polymerization, as shown in Figure 4(Ac) 6.486 nm, Figure 4(Ag) 8.263 nm, Figure 4(Ad) 7.039 nm, and Figure 4(Ah) 7.694 nm. These results explained that amine polymeric thin film had no effect on the roughness of PCL/β-TCP scaffold surfaces. 

The PCL film surfaces were evaluated by FTIR analysis in order to investigate the changes in functional organic groups in the PCL film after adding β-TCP and amine plasma-polymerization. The FTIR spectra of the PCL, PCL/β-TCP, PCL/β-TCP treated AA plasma, and PCL/β-TCP treated DACH plasma are presented in Figure 5. The spectrum of PCL shows a peak around 2945 cm^−1^ and a very sharp signal at 1723 cm^−1^, corresponding to CH and C=O groups, respectively. A band at 1293 cm^−1^ is known to be associated with the backbone C–C and C–O groups of PCL spectrum, which characterize the structure of the PCL scaffold [28]. The typical spectrum of β-TCP can be observed 570 and 630 cm^−1^ attributed to the bending vibration mode of the O–P–O group. The 969 and 947 cm^−1^ peaks were caused by the symmetric stretching mode of the P–O bond, and the peaks at 1100 and 1040 cm^−1^ are assigned to the antisymmetric stretching mode of the P–O bond [29]. In the case of PCL/β-TCP film, the peaks at 570 and 630 cm^−1^ correspond to the absorption bands of O–P–O group, which is an identification of incorporation of the β-TCP in the PCL film.

In the spectrum of PCL/β-TCP films, amines adsorption bands were not observed. However, amine plasma-polymerized PCL/β-TCP films shows several absorption bands characteristic of the presence of primary amine peak at 1630 cm^−1^ and/or secondary amines groups at 3380 cm^−1^. These results demonstrate that amine monomers have been successfully polymerized onto the PCL/β-TCP film surface. The PCL/β-TCP DACH surface showed the highest intensity bands for the amine groups, as compared to the other surfaces.

The water contact angles on the samples are shown in Figure 6. The contact angle measurement was performed by an image analysis system that calculates both left and right contact angles of the droplet shape. The contact angles of distilled water on the pristine PCL and PCL/β-TCP films were approximately 67.28° ± 0.39° and 70.35° ± 7.53°, respectively. After AA and DACH plasma-polymerization, the PCL/β-TCP surfaces showed the contact angles of approximately 16.85° ± 3.00° and 19.60° ± 3.21°, respectively. After plasma-polymerization, the contact angles of PCL and PCL/β-TCP films were decreased. Generally, surface wettability of scaffold is increased by the introduction of the polar functional groups to the mostly inert surfaces of polymers [17].

Universal Testing Machine (UTM) was used to test the compressive strength of 3D scaffold samples. Figure 7 shows the typical stress–strain curves of all 3D scaffolds tested. The strain–stress curves of all 3D scaffold showed a similar trend and it tended to be slightly deceased when 10 wt% β-TCP was added. However, no significant difference was found.

### 3.2. In Vitro Pre-Osteoblast Evaluation

In vitro studies were performed to evaluate the effect the incorporation of β-TCP and amine plasma-polymerization on the biological response of MC3T3-E1 cells grown on PCL and PCL/β-TCP scaffolds. MTT assay was used to evaluate the MC3T3-E1 cell proliferation on the PCL and PCL/β-TCP scaffolds before and after amine plasma-polymerization cultured for 1, 3, and 5 days, as shown in Figure 8. The amine plasma-polymerization treated scaffolds showed higher MC3T3-E1 proliferation than pristine PCL and PCL/β-TCP samples. The DACH plasma-polymerization treated PCL/β-TCP showed the highest cell proliferation among all samples. There is no difference between pristine PCL and PCL/β-TCP scaffolds, and PCL/β-TCP scaffolds treated with AA and DACH plasma-polymerization at all culture points show more dominant cell proliferation than scaffolds without any treatment. On day one and three of cell culture, DACH plasma treated PCL/β-TCP scaffold showed higher cell proliferation than AA plasma treated PCL/β-TCP scaffold, and similar cell proliferation pattern was observed on day five. These results suggest that cell proliferation is significantly affected by hydrophilicity and amine group of the scaffold surface. All the group-treated amine plasma-polymerization showed statistically significant differences between pristine PCL and PCL/β-TCP scaffolds. 

### 3.3. Live and Dead Cell 

The live/dead assay, a two-part dye staining live cells green and dead cells red, was used to evaluate the cell viability on PCL scaffolds and PCL/β-TCP scaffolds before and after plasma-polymerization after two days culture. After 2 days, cells were not observed on the top layer of the PCL and PCL/β-TCP scaffolds, as shown in Figure 9. On the other hand, PCL/β-TCP scaffolds treated amine plasma-polymerization were covered with MC3T3-E1 cells on the topmost layer. This result indicates that amino-functionalized scaffold surfaces provide a chemically modified surface site for the easy adherence of pre-osteoblast. 

### 3.4. Cell Focal Adhesion

The immunofluorescence images of focal adhesion of MC3T3-E1 cells cultured for 5 h on scaffolds is presented in Figure 10. Figure 10A shows the fluorescence images of vinculin protein expression in MC3T3-E1 cells on four different scaffolds, and paxillin protein expression was shown in Figure 10B. The actin cytoskeleton shows that the cells attached on the plasma-polymerized PCL/β-TCP surfaces were more elongated and flattened than those on untreated PCL and PCL/β-TCP surfaces. In addition, plasma-polymerized PCL/β-TCP surfaces had more observed vinculin and paxillin protein expression than pristine PCL and PCL/β-TCP surfaces. 

### 3.5. ALP Activity 

The MC3T3-E1 cell differentiation on the PCL and PCL/β-TCP scaffolds before and after amine plasma-polymerization were evaluated by measuring ALP activity and ALP staining after 7 and 14 days. Figure 11A shows the results of the ALP activity assay. On the 14th day of differentiation, amine plasma-polymerized PCL/β-TCP scaffolds show a significant difference compared to untreated PCL and PCL/β-TCP scaffolds.

Similar to the results of ALP activity, differentiation was enhanced at day 14 compared to day 7 in all samples. Notably, ALP staining on PCL/β-TCP AA scaffolds was significantly increased compared to the control group on the 14th day of differentiation (Figure 11B).

### 3.6. Alizarin Red Staining

Alizarin red staining was performed on days 7 and 14 of differentiation to evaluate the bone mineralization of MC3T3-E1 cells in various scaffolds (Figure 12A). The calcium deposition is analyzed qualitatively using the Alizarin Red staining method (Figure 12B). On the 7th day of differentiation, the DACH plasma-polymerization treated PCL/β-TCP scaffold group was stained a stronger red color compared to the other groups. This means that the calcium deposition of PCL/β-TCP scaffolds treated with DACH plasma-polymerization was highest compared to other groups. All the groups showed a statistically significant difference (* *p* < 0.05). On the 14th day of differentiation, similar to the 7th day of differentiation, the DACH showed the highest bone mineralization, and there was little difference between the rest of the groups.

### 3.7. Western Blot

Figure 13 shows the expression of RUNX2, ALP, OCN, and OPN marker proteins determined by Western blot analysis after osteogenic differentiation for 7 and 14 days. In 7 days, the expression level of the ALP protein was significantly increased in amine plasma-polymerization groups compared to other groups, and there was no difference in the expression level of Runx2 and OPN protein in all groups. However, closer to 14 days, expression levels of the all proteins in amine plasma-polymerization groups were increased compared to other groups. The results confirmed that the expressions of ALP, OPN, and Runx2 in differentiation-induced MC3T3-E1 were significantly elevated in PCL/β-TCP scaffold treated amine plasma-polymerization.

## 4. Discussion

Three-dimensional printing based on additive manufacturing is widely used as a technology that can manufacture ideal 3D scaffolds for bone tissue engineering application. One of the advantages of 3D printing is its ability to control the scaffold architecture’s internal structure. In present work, optimal design of 3D scaffold architecture’s internal structure was determined through the results of our previous studies [30,31]. For example, a 3D scaffold used in this work fabricated a 0°/45° strut layout pattern and 300 μm pore size. Among the 3D printing, fused deposition modeling (FDM) is a 3D printing process that uses thermoplastic polymers such as polylactic acid (PLA), poly(lactic-co-glycolic acid) (PLGA), and polycaprolactone (PCL). Notably, the PCL is more slowly absorbed in human body than PLA or PLGA, and it is extensively used as scaffold material due to its toughness and biocompatibility [32,33]. 

Β-tricalcium phosphate (β-TCP) is a bioresorbable ceramic that is rapidly degraded into calcium and phosphate in the body, having a similar mineral composition to bone and teeth. Therefore, although it is considered as excellent material in bone tissue engineering, β-TCP does not have sufficient mechanical strength because it exists in powder form [34,35]. Consequently, in order to fabricate scaffolds using β-TCP, an additional process is needed to form composites with other biopolymers [36,37]. Until now, numerous studies have been conducted to improve the bone regeneration of the 3D PCL/β-TCP scaffolds for bone tissue engineering application [38,39,40,41]. 

Shin et al. reported on the influence of calcium phosphate on osteogenic differentiation of human mesenchymal stem cells, demonstrating an increasing ALP activity of scaffolds with calcium phosphate compare to PCL only [42]. 

Zhang et al. found that extrusion-based 3D printing of PCL and β-TCP encapsulated with ICA composite scaffolds, promoted differentiation of BMSCs in vitro [43]. 

Plasma surface modification (PSM) has been widely used to change the surface characteristics of biopolymers to achieve a better biocompatibility without altering the bulk properties [44]. Among the PSM, plasma-polymerization can provide a large number of functional groups on the surface because the gradient of their density is homogeneous [21,45]. In addition, plasma-polymerization has been demonstrated, e.g., to enhance the biocompatibility of vascular stents [46], favor cell proliferation [47], and immobilize biomolecules [48,49].

In this study, we performed research on how the amino-functionalized surface through amine plasma-polymerization on PCL/β-TCP scaffolds affect the biological response of pre-osteoblasts for bone regeneration. In doing so, we compared the pre-osteoblast biological activity on pristine 3D PCL/β-TCP scaffolds, and on 3D PCL/β-TCP scaffolds modified with amine plasma-polymerization. 

In the present study, the contact angle of pristine PCL/β-TCP films was approximately 70.35° ± 7.53° and it showed hydrophobic property; however, after AA and DACH plasma-polymerization, the contact angles of the PCL/β-TCP surface changed to approximately 16.85° ± 3.00° and 19.60° ± 3.21°, respectively (Figure 6). 

The contact angle results revealed that amine modification of the PCL/β-TCP surface, change surface chemical compositions and also enhance the surface wettability. The surface wettability is controlled mostly by the charge, polarizability, and polarity of the surface functional groups [50]. According to previous studies, surface wettability has an important role in influencing the biological response of a biomaterial, and cells are more likely to adhere to hydrophilic surfaces [51]. For example, Wei et al. found that the adhesion of osteoblasts decreased when the contact angle increased from 0° to 106° [52]. 

Eriksson et al. were performed the implantation of hydrophilic and hydrophobic titanium discs in rat tibia. Bone formation in implants with highly hydrophilic surfaces is enhanced compared to that of hydrophobic surface [53].

In Figure 10, amine plasma-polymerized PCL/β-TCP surface was more observed the vinculin and paxillin expression compared to pristine PCL/β-TCP surfaces. Vinculin frequently links adhesion receptors (e.g., integrins) to the contractile actin-myosin cytoskeleton by binding either Talin through its amino-terminal globular head domain [54], or paxillin through its rod-like tail domain [55].

Amine surface modification is known to improve the cell adhesion by enhancing integrin binding, which is required for osteoblastic differentiation [56,57,58]. In detail, cell adhesion is mainly modulated by the binding of cellular integrins and adhesive proteins, such as fibronectin, in the extra-cellular matrix (ECM). The positive charges of amines cause an increase in the density of fibronectin and change its conformation [59]. Increasing the density of fibronectin enhances cell adhesion by increasing the binding to integrins [60,61], and triggers rapid phosphorylation of focal adhesion-associated tyrosine kinase (FAK) [62]. 

Our study also demonstrated that the amine plasma-polymerization enhanced cellular proliferation and osteogenic differentiation (Figure 8 and Figure 11), and bone mineralization was also enhanced (Figure 12). 

From these results, the presence of amine groups on PCL/β-TCP scaffold surfaces triggered extracellular signal-regulated kinase (ERK)/mitogen-activated protein kinase (MAPK) signaling to upregulate Runt-related transcription factor 2 (Runx2), which is a master regulator of osteoblastic differentiation [63,64,65,66].

The findings of this study revealed that the plasma-polymerized amine polymeric thin film on PCL/β-TCP scaffold surfaces contributed to the cell adhesion, proliferation, and osteoblastic differentiation capability.

## 5. Conclusions

The 3D PCL and PCL/β-TCP scaffolds with interconnected pores were successfully fabricated using FDM 3D printing. A three-dimensional PCL/β-TCP scaffold was modified by amine plasma-polymerization using AA and DACH monomers in order to improve the MC3T3-E1 cell bioactivity in vitro. After plasma-polymerization, the hydrophilicity of the 3D PCL/β-TCP scaffold surface was significantly increased and surface roughness was not changed. In addition, amine plasma-polymerization was seen to positively influence cell behaviors, such as focal adhesion, proliferation, and osteogenic differentiation. Notably, DACH plasma-treated 3D scaffold showed the highest bioactivity compared to other 3D scaffold samples. Based on these results, amine plasma-polymerization is an effective technique for modulating the osteoblast biological behaviors of the 3D-printed PCL/bioceramic scaffolds for bone tissue engineering application.

## Figures and Tables

**Figure 1 materials-15-00366-f001:**
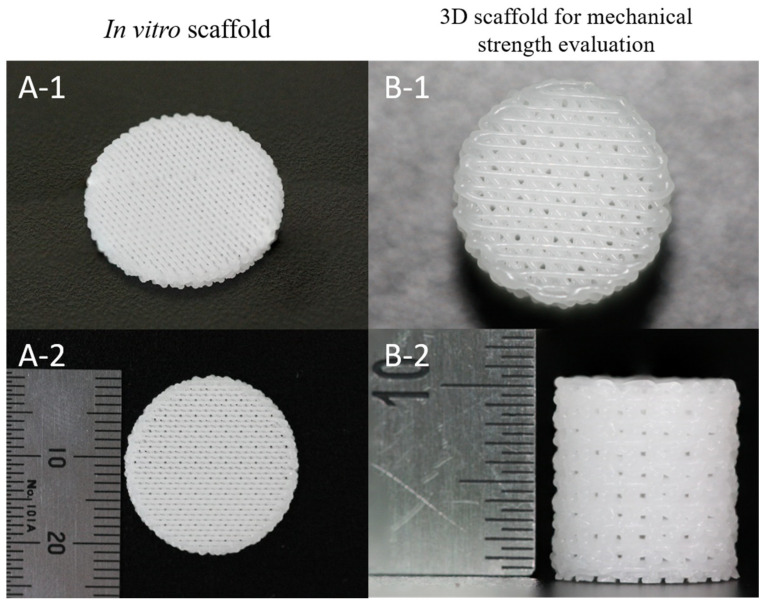
Scaffold photographs used in this study: (**A-1**) plane view image, (**A-2**) side view image of 3D scaffold for in vitro test; (**B-1**) plane view image, (**B-2**) side view image of 3D scaffold for mechanical strength evaluation ((**A-1**): diameter = 20 mm, (**A-2**): height = 1.2 mm, (**B-1**): diameter = 10 mm, (**B-2**): height = 10 mm).

**Figure 2 materials-15-00366-f002:**
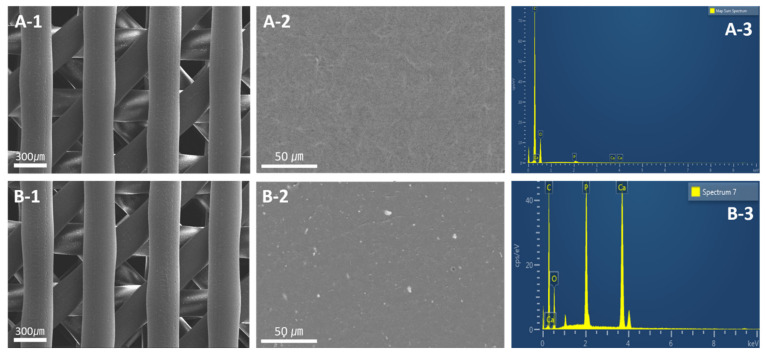
FE-SEM images and EDS analysis of the (**A-1**) 3D PCL scaffold, (**A-2**) PCL scaffold surface, and (**A-3**) EDS spectrum of PCL scaffold; (**B-1**) 3D PCL/β-TCP scaffold, (**B-2**) PCL/β-TCP scaffold surface, and (**B-3**) EDS spectrum of PCL/β-TCP scaffold.

**Figure 3 materials-15-00366-f003:**
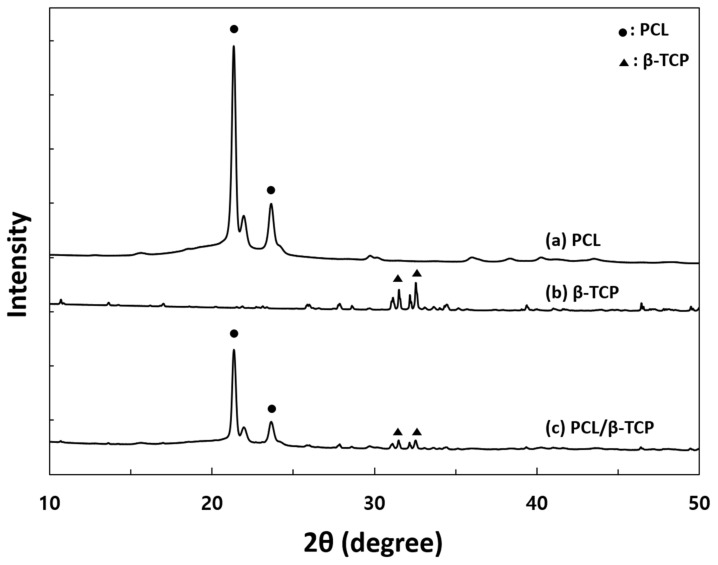
XRD patterns of: (a) PCL scaffold, (b) β-TCP powder, and (c) PCL/β-TCP scaffolds.

**Figure 4 materials-15-00366-f004:**
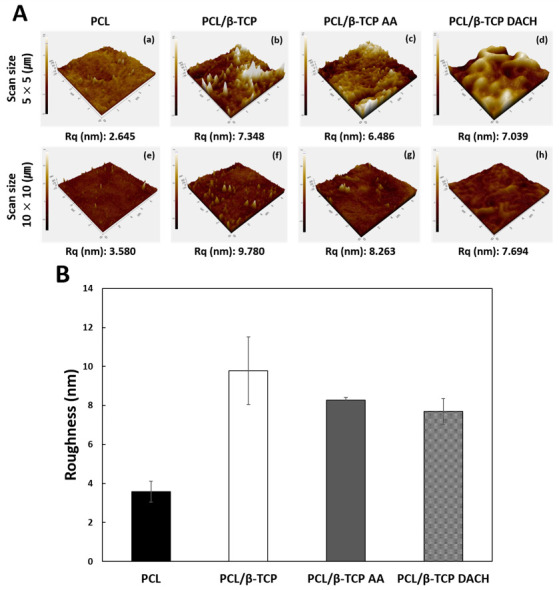
(**A**) Representative AFM 3D-topographical images of the (**a**) PCL scaffold, (**b**) PCL/β-TCP scaffold, (**c**) PCL/β-TCP scaffold treated by AA plasma for 1 min, (**d**) PCL/β-TCP scaffold treated by DACH plasma for 1 min. For (**a**–**d**), the scan area = 5 × 5 μm. For (**e**–**h**), the scan area = 10 × 10 μm. (**B**) Roughness values of the PCL film, PCL/β-TCP film, PCL/β-TCP AA film, and PCL/β-TCP DACH film (*n* = 3).

**Figure 5 materials-15-00366-f005:**
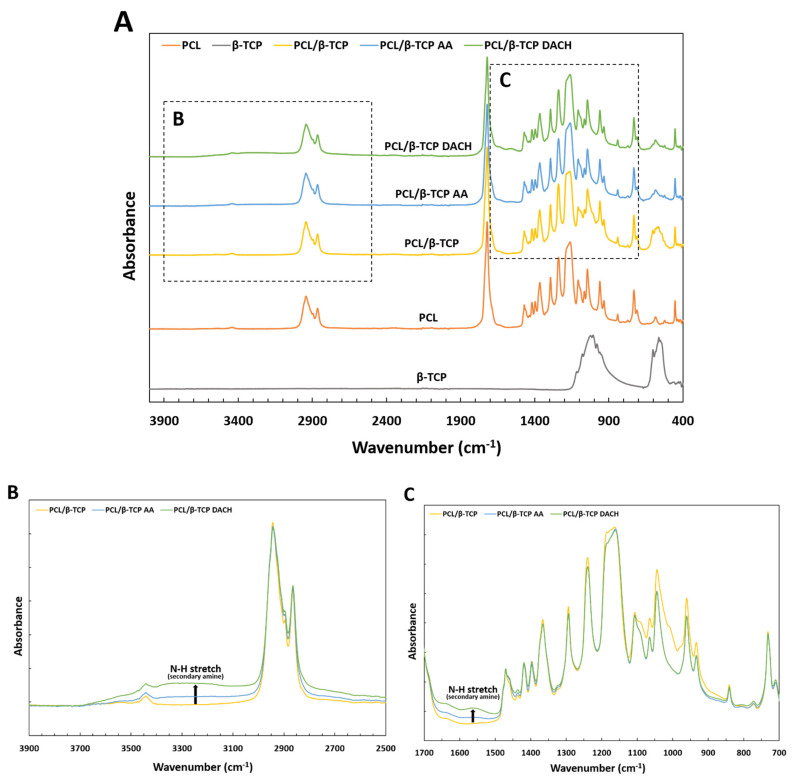
(**A**) FTIR spectra of β-TCP, PCL PCL/β-TCP, PCL/β-TCP AA, and PCL/β-TCP DACH. (**B**) FTIR spectra of PCL/β-TCP, PCL/β-TCP treated AA plasma, and PCL/β-TCP treated DACH plasma at wavenumber of 3900–2500 cm^−1^. (**C**) FTIR spectra of PCL/β-TCP, PCL/β-TCP treated AA plasma, and PCL/β-TCP treated DACH plasma at wavenumber of 1700–700 cm^−1^.

**Figure 6 materials-15-00366-f006:**
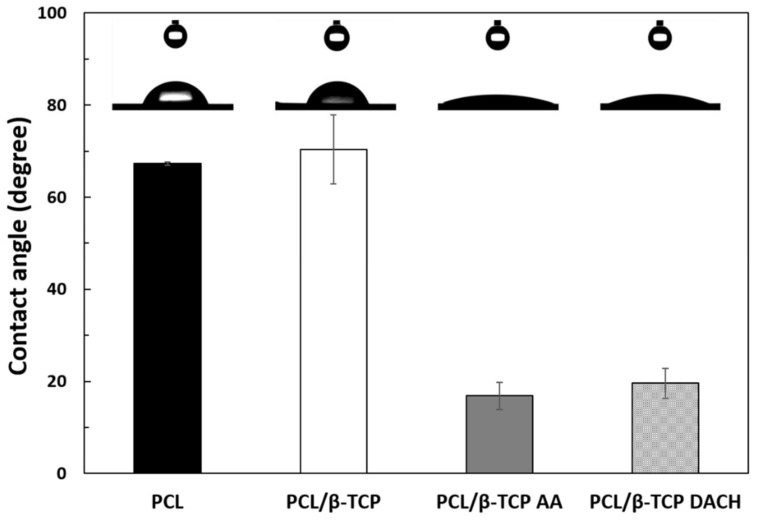
Water contact angles of PCL, PCL/β-TCP, PCL/β-TCP AA and PCL/β-TCP DACH films.

**Figure 7 materials-15-00366-f007:**
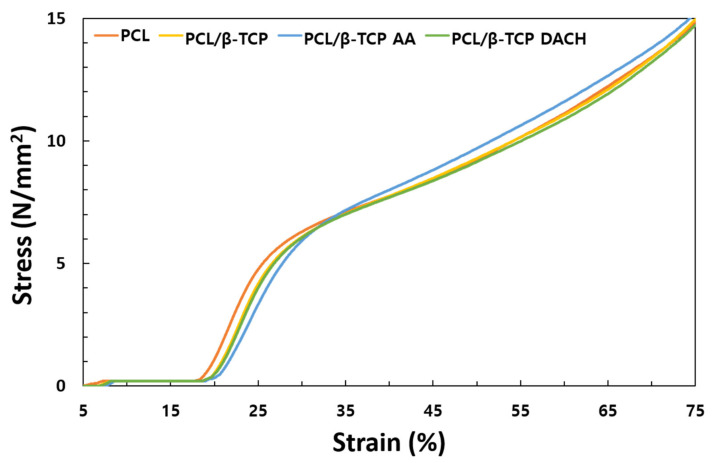
Stress–strain curves obtained for 3D PCL and PCL/β-TCP scaffolds with a 0/45° lay-down pattern compressed at cross-head speed of 0.5 mm/min.

**Figure 8 materials-15-00366-f008:**
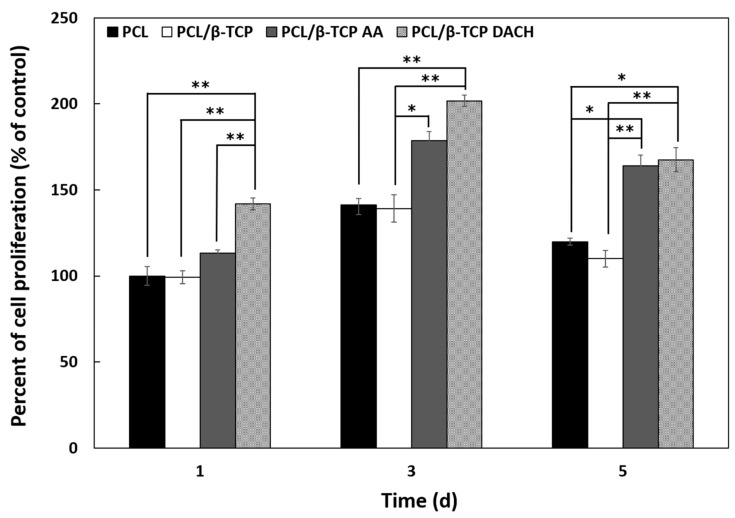
Evaluation of cell proliferation on 3D scaffolds: Cell growing within PCL, PCL/β-TCP, PCL/β-TCP AA, and PCL/β-TCP DACH scaffolds for 1, 3, and 5 days, as determined by the MTT assay. Cell proliferation was found significantly higher in DACH plasma treated 3D PCL/β-TCP scaffolds as compared to the pristine PCL scaffold on one day (*n* = 3, ** *p* < 0.01 and * *p* < 0.05 compared to values shown by PCL scaffolds on one day). Statistical analysis was carried out using One-way ANOVA followed by post hoc Tukey test.

**Figure 9 materials-15-00366-f009:**
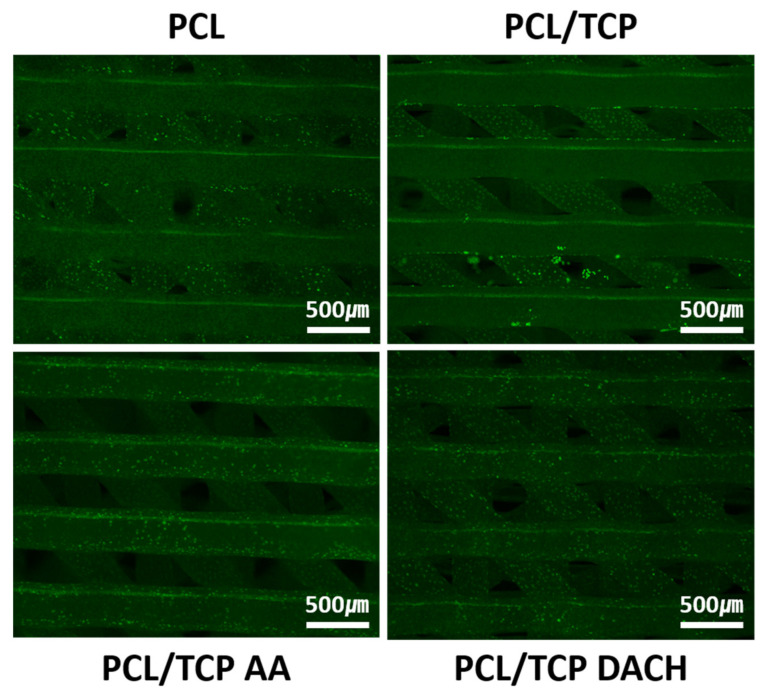
Live and Dead cell staining images of MC3T3-E1 cells cultured on the PCL scaffolds and PCL/β-TCP scaffolds before and after allylamine and DACH plasma polymerization after 2 days. Live cells are indicated by green and dead cells are red (40×).

**Figure 10 materials-15-00366-f010:**
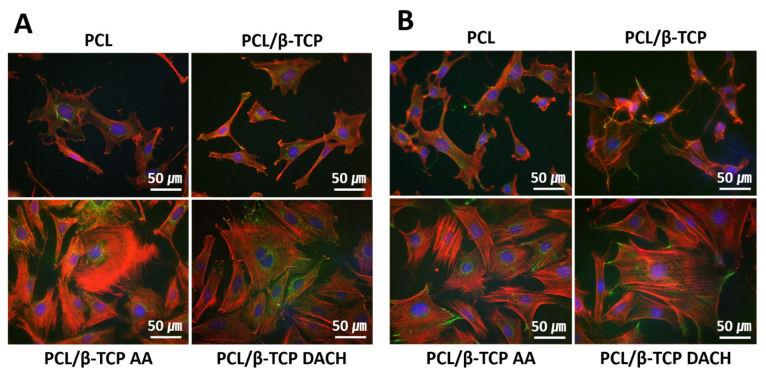
Fluorescence microscopy images of MC3T3-E1 cells adhered to each scaffold. After 5 h cell seeding, samples were fixed and processed for immunofluorescence using an Alexa-Fluor 488 fluorescent antibody to detect vinculin and paxillin (green), Rhodamine-phalloidin to label actin (red), and DAPI to label nucleic acids in the nucleus (blue). Each image is representative of both groups (**A**) vinculin and (**B**) paxillin (400×).

**Figure 11 materials-15-00366-f011:**
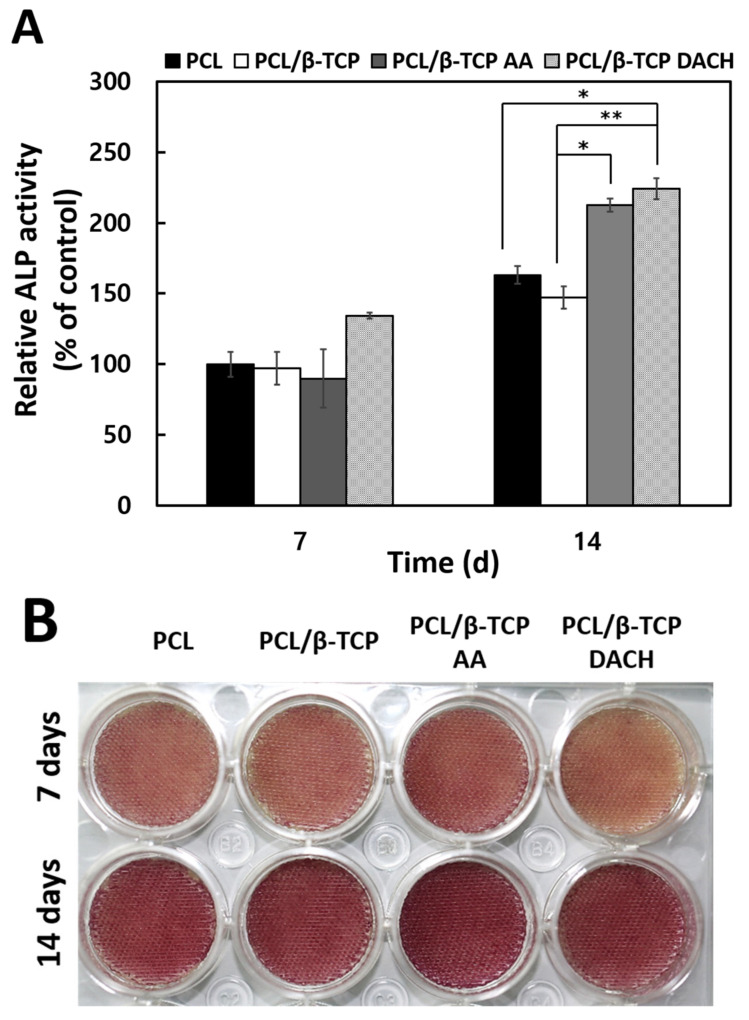
Osteogenic differentiation study: (**A**) ALP activity of MC3T3-E1 cells cultured for 7 and 14 days on the 3D pristine PCL scaffold and PCL/β-TCP scaffold before and after AA and DACH plasma polymerization (*n* = 3, * *p* < 0.05, ** *p* < 0.01 compared with the PCL scaffolds on 7 days). Statistical analysis was carried out using One-way ANOVA followed by post-hoc Tukey test. (**B**) ALP staining photographs for the stained pristine PCL, PCL/β-TCP, and amine plasma polymerized PCL/β-TCP scaffolds cultured with cells for 7 and 14 days.

**Figure 12 materials-15-00366-f012:**
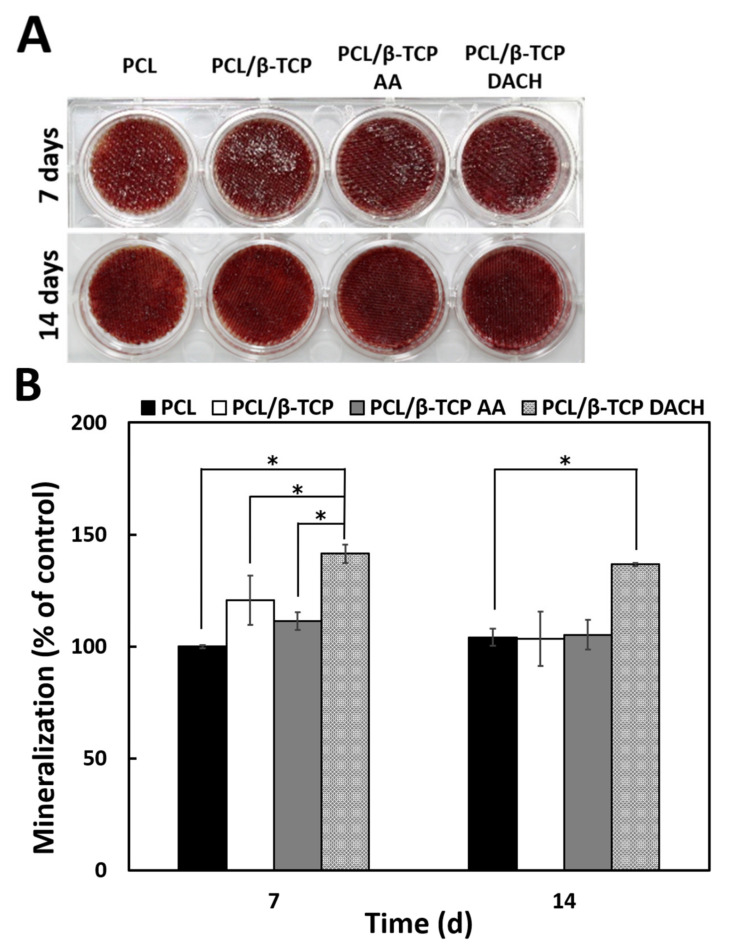
(**A**) Photographs of Alizarin red staining for the PCL, PCL/β-TCP, and amine plasma polymerized PCL/β-TCP scaffolds cultured with cells for 7 and 14 days and (**B**) Quantitative Ca deposition data from Alizarin red staining experiments on MC3T3-E1 cells cultured for 7 and 14 days on the 3D pristine PCL scaffold and PCL/TCP scaffold before and after allylamine and DACH plasma polarization. (*n* = 3, * *p* < 0.05 compared with the value of PCL scaffolds on 7 day). Statistical analysis was carried out using One-way ANOVA followed by post hoc Tukey test.

**Figure 13 materials-15-00366-f013:**
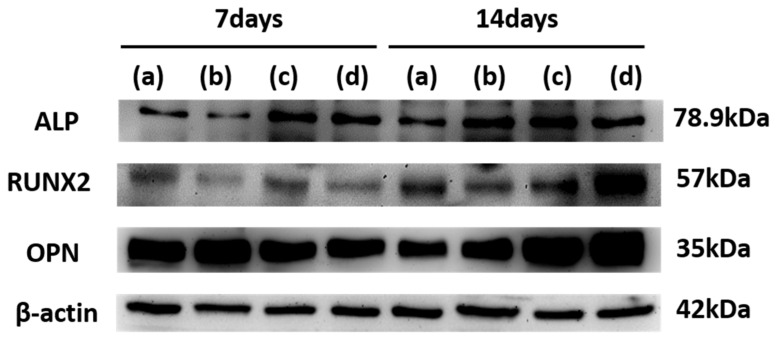
Western blot analysis of (a) PCL, (b) PCL/β-TCP, (c) PCL/β-TCP treated AA plasma, (d) PCL/β-TCP treated DACH plasma. The expression of RUNX2, ALP and OPN proteins were determined by Western blot analysis after osteogenic differentiation for 7 and 14 days. Runx2, Runt-related transcription factor 2; ALP, Alkaline phosphatase; OPN, Osteopontin.

**Table 1 materials-15-00366-t001:** Plasma-polymerization conditions.

Pre-Treatment and Post-Treatment	Gas Flow Rate (sccm)	Pressure (mTorr)	Power (W)	Time (s)
Argon gas	20	100	30	10
**Plasma-Polymerization**	**Precursor**	**Pressure (mTorr)**	**Power (W)**	**Time (s)**
AA	Allylamine	30	60	60
DACH	1,2-diaminocyclohexane	10	80	60

## Data Availability

Data sharing not applicable.

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
