# Peer review of "Amine Plasma-Polymerization of 3D Polycaprolactone/β-Tricalcium Phosphate Scaffold to Improving Osteogenic Differentiation In Vitro"

_materials, 2022, doi:10.3390/ma15010366_

Round 1

Reviewer 1 Report

The authors present a study on the preparation and properties of a poly(ε-caprolactone)/β-tricalcium phosphate (β-TCP) scaffold modified by amine plasma-polymerization fabricated using fused deposition modeling 3D printing. This study is focused on demonstrating the positive effect of plasma-polymerization on cell bioactivity in vitro. Although the technical approach is interesting, some of the measurements are missing some discussions. I consider the manuscript suitable for publication only after appropriately addressing the suggestions below:

  1. In the introduction, the authors mentioned the advantages of using 3D printing and highlight the benefits of 3D printing on the scaffold architecture's internal structure. However, the internal structure of the printed scaffold is not fully described in the results, the porosity and pore size are not discussed. The SEM images reported in Figure 2 show a 250 um pore size. Could the authors explain their approach on designing their optimal architecture for bone regeneration?
  2. Is this optimal design stated by the authors also meeting the degradability requirements? Could the authors discuss this aspect?
  3. The authors reported some contact angle measurements taken on pristine PCL/β-TCP films and after plasma-polymerization. Could the authors discuss those values regarding the cell adhesion and viability?

Author Response

Thank you very much your good suggestions.

  1. In the introduction, the authors mentioned the advantages of using 3D printing and highlight the benefits of 3D printing on the scaffold architecture's internal structure. However, the internal structure of the printed scaffold is not fully described in the results, the porosity and pore size are not discussed. The SEM images reported in Figure 2 show a 250 um pore size. Could the authors explain their approach on designing their optimal architecture for bone regeneration?

Answer) Thank you for your good comments. According to your suggestion, “one of the advantages of 3D printing is possible to controllable the scaffold architecture's internal structure. In present work, optimal design of 3D scaffold architecture’s internal structure was determined through the results of our previous studies [30,31]. For example, 3D scaffold used in this work fabricated 0°/45° strut layout pattern and 300 μm pore size.” this paragraphs and references have been inserted in discussion.

[30]  Kook, M.S.; Roh, H.S.; Kim, B.H. Effect of oxygen plasma etching on pore size-controlled 3D polycaprolactone scaffolds for enhancing the early new bone formation in rabbit calvaria. Dent. Mater. J. 2018, 37, 599–610. 

[31]   Roh, H.S.; Jung, S.C.; Kook, M.S.; Kim, B.H. In vitro study of 3D PLGA/n-HAp/β-TCP composite scaffolds with etched oxygen plasma surface modification in bone tissue engineering. Appl. Surf. Sci. 2016, 388, 321–330.

----------------------------------------------------------------------------------------------------------------

  1. Is this optimal design stated by the authors also meeting the degradability requirements? Could the authors discuss this aspect?

Answer) Thank you for your good comments. When considering the optimal scaffold design in terms of bone regeneration, degradability is thought to be more important than scaffold architecture, but rather the properties of the material itself composing the scaffold. Therefore, degradability was not stated, and it not even mentioned in the discussion.      

 --------------------------------------------------------------------------------------------------------------

  1. The authors reported some contact angle measurements taken on pristine PCL/β-TCP films and after plasma-polymerization. Could the authors discuss those values regarding the cell adhesion and viability?

Answer) Thank you for your good comments.  As shown in Figure 6. contact angles of PCL/β-TCP films and amine plasma-polymerized PCL/β-TCP films show 70.35° ± 7.53° and 16.85° ± 3.00°, respectively. In general, cells are more likely to adhere to hydrophilic surfaces. In detail, Wei et al reported that initial cell attachment was enhanced when the contact angle decreased from 106° to 0°. This content and references were already mentioned at line 605 – line 610 and References [49, 50].

Reviewer 2 Report

The manuscript is relevant for publication in the Materials. The work reports the study is to investigate the surface characterization and preosteoblast biological behaviors on the 3D poly(ε-caprolactone)/β-tricalcium phosphate scaffold modified by amine plasma-polymerization. The work is well written, contextualized, and very relevant. Regarding the results and discussion, the experimental observations are well discussed. To accept the article, it is necessary to make some changes/suggestions below:

1- Update bibliographic references.

2- Recommend reducing the introduction.

3- What is the pressure of the experiment for the formation of scaffolding? Topic 2.2.

4- What is the purpose of platinum coating? It is usually gold (FE-SEM).

5- FTIR carried out with KBr? Must mentioned in the information. FTIR spectrum has no peak (use bands).

6- “Two days in CO2 incubator. “ line 262

7- Topic 2.6.5: Reagents information should be stated in topic 2.1.

8- “Generally, the surface wettability of scaffold is increased by the introduction of the polar functional groups to the mostly inert surfaces of polymers.” Reference.

9- “These results suggest that cell proliferation is significantly affected by the hydrophilicity and amine group of the scaffold surface.” What is the justification? Topic 3.2.

Author Response

Thank you very much your good suggestions.

  1. Update bibliographic references.

Answer) Thank you for your good comment. According to your suggestion, bibliographic references were updated. Please see revised references.

  1. Recommend reducing the introduction.

Answer) Thank you for your good comment. According to your suggestion, introduction was reduced. Please see revised introduction.

  1. What is the pressure of the experiment for the formation of scaffolding? Topic 2.2.

Answer) Thank you for your good point out. According to your suggestion, “PCL and PCL/β-TCP mixtures were ejected through heated nozzle and the feed rate was set to 240 mm/min” has been revised to “PCL and PCL/β-TCP mixtures were ejected through heated nozzle of compressed dry air at a pressure of 640 kPa, and the feed rate was set to 240 mm/min”.

  1. What is the purpose of platinum coating? It is usually gold (FE-SEM).

Answer) Thank you for your good point out. We performed platinum coating to observe the FESEM images. We have a mistake in use the term. According to your suggestion, platinum coating was corrected to gold coating.

  1. FTIR carried out with KBr? Must mentioned in the information. FTIR spectrum has no peak (use bands).

Answer) Thank you for your good comment. KBr was not used in this experiment due to it is difficult to blending with PCL polymer.

  1. “Two days in CO2 incubator. “ line 262

Answer) Thank you for your good point out. CO2 was corrected to CO2.

  1. Topic 2.6.5: Reagents information should be stated in topic 2.1.

Answer) Thank you for your good comment. Reagents information was moved to Topic 2.1.

Please see 2.1. Materials.

  1. “Generally, the surface wettability of scaffold is increased by the introduction of the polar functional groups to the mostly inert surfaces of polymers.” Reference.

Answer) Thank you for your good comment. Reference [17] was added to this paragraph.  

  1. Desmet, T.; Morent, R.; De Geyter, N.; Leys, C.; Schacht, E.; Dubruel, P. Nonthermal Plasma Technology as a Versatile Strategy for Polymeric Biomaterials Surface Modification: A Review. Biomacromolecules 2009, 10, 2351–2378.

  1. “These results suggest that cell proliferation is significantly affected by the hydrophilicity and amine group of the scaffold surface.” What is the justification? Topic 3.2.

Answer) Thank you for your good comment. This paragraph you mentioned is not need to be mentioned in the results, it will be mentioned in the discussion. Therefore, “These results suggest that cell proliferation is significantly affected by hydrophilicity and amine group of the scaffold surface.” Has been removed in the results. This has been re-mentioned in discussion.
